# From Queries to Clones: A Systematic Study of Encoder Stealing Attacks

## Abstract

Model stealing is a growing threat for online ML services. Encoder models are more vulnerable to stealing than classifiers because their high-dimensional embeddings reveal substantially richer information than class logits. This risk is amplified by Encoder-as-a-Service platforms that expose foundation encoders. Prior work mostly studied encoder stealing attacks in isolation with inconsistent setups, so practical trade-offs and failure modes across attacks remain unclear. We present a comprehensive benchmark and comparative study of a representative set of encoder stealing attacks on two widely used encoders, CLIP and DINO. We consider three threat scenarios with increasing realism: (i) the attacker has access to the victim's training data, (ii) the attacker knows the victim's training distribution and uses disjoint data, and (iii) the attacker has no reliable knowledge of the victim's training distribution. We also evaluate a novel setting where the attacker uses queries from multiple datasets to steal a more generalizable surrogate. Finally, we vary data and query budgets, surrogate capacity, and resource constraints to understand practical attack scenarios. Across our settings, we observe that high utility on the stealing distribution does not necessarily translate to high utility on the victim's training distribution under shift. We find that contrastive objectives with strong augmentations are the most reliable, conventional methods can be brittle, and prototype alignment is query-efficient but shifts cost to local compute and memory. In our experiments, mixed-source queries reveal a data density-diversity trade-off, and DINO is consistently easier to steal than CLIP, with cross-modal text guidance partially narrowing the gap. Overall, our results map practical attack operating points and highlight vulnerabilities relevant to the foundation model era.

## 1 Introduction

The deployment of machine learning models through online services has introduced significant security and privacy risks, with model stealing (or extraction) emerging as a new threat (Tramèr et al., 2016). In such attacks, an adversary repeatedly queries a victim model and uses its outputs to train a surrogate that approximates the victim's behavior. Extraction attacks compromise the providers' intellectual property and enable a range of downstream threats, including adversarial examples and membership inference (Shokri et al., 2017; Papernot et al., 2017; Fredrikson et al., 2015). Early work showed that classifiers exposed via prediction APIs can be effectively reproduced using only labels or logits (Tramèr et al., 2016; Orekondy et al., 2019). More recent studies have shifted focus to encoders that map inputs to high-dimensional representations reused across many downstream tasks. Vision encoders present especially attractive targets because their embeddings are richer and more informative than task-specific predictions, and a single stolen encoder can support a wide range of tasks. The rise of large-scale foundation encoders such as CLIP (Radford et al., 2021) and DINO (Caron et al., 2021) further amplifies these risks. Trained on massive datasets with substantial compute budgets and reused as backbones across many downstream tasks (Rao et al., 2022; Sharma et al., 2025; Lüddecke & Ecker, 2022; Han & Lim, 2024; Barsellotti et al., 2025), these models are increasingly deployed through Encoder-as-a-Service (EaaS) embedding APIs. Since these encoders concentrate significant capability and investment behind API interface, their theft would allow adversaries to obtain powerful, general-purpose representations that would otherwise require substantial resources and expertise.

Recent work (Dziedzic et al., 2022; Liu et al., 2022; Sha et al., 2023; Wu et al., 2024; Zhang et al., 2024) has begun to explore encoder stealing attacks in this setting, but the literature remains narrow in scope. Most existing studies target CNN backbones or self-supervised encoders trained on small-scale vision datasets, often under favorable conditions where the attacker has access to data drawn from the same or a highly similar distribution as the victim (Dziedzic et al., 2022; Sha et al., 2023; Liu et al., 2022; Wu et al., 2024). As a result, we still lack a systematic understanding of how encoder stealing behaves when we vary: (i) the attacker's knowledge of the victim's data distribution, (ii) whether the victim is a task-specific or a large foundation encoder, (iii) the choice of stealing objective, and (iv) the composition of the queries issued to the victim. Moreover, while several works report downstream accuracies of stolen models, the connection between *representation-level alignment* of embeddings and downstream performance remains poorly understood.

This paper addresses these gaps by introducing a systematic benchmark of encoder-stealing attacks on vision encoders under realistic black-box settings. We consider both task-specific self-supervised encoders and foundation models (CLIP and DINO) accessed only through embedding outputs, with no knowledge of the victim's architecture, data, or training details. Within this setting, we evaluate three representative encoder-stealing methods, Conventional (Dziedzic et al., 2022), Cont-Steal (Sha et al., 2023), and Representation Distribution Alignment (RDA) (Wu et al., 2024), and analyze how their performance depends on the attacker's data and query strategies. We distinguish three attacker knowledge regimes: *dataset-aware* (queries match the victim's training data), *distribution-aware* (queries come from a similar but not identical distribution), and *data-agnostic* (only generic, unrelated images). We further extend the analysis to mixed dataset attacks that combine queries from multiple data distributions.

In addition to direct vision-to-vision stealing, we propose targeting multi-modal foundation models. In this setting, the attacker does not rely solely on aligning the surrogate vision encoder's embeddings with those of the victim vision encoder; they also use the victim's text encoder for multi-modal alignment. This allows us to investigate if grounding the stealing process in the victim's text-image alignment leads to more robust and versatile surrogate models. A distinctive feature of our benchmark is that it jointly evaluates *task-specific* and *foundation* victim encoders under a unified threat model. This allows us to characterize not only how well an attacker can reproduce a given encoder, but also to compare the inherent *stealability* of different encoder families and architectures. Our contributions are as follows:

1. We characterize how stealing performance degrades across dataset-aware, distribution-aware, and data-agnostic attacker regimes.

2. We reveal systematic differences in stealability across task-specific and foundation encoders, and vulnerabilities in CLIP and DINO.

3. We study mixed-source query strategies and provide a data density-diversity trade-off for stealing a more generalizable surrogate encoder.

4. We study a cross-modal stealing strategy for CLIP that leverages the victim's text encoder to improve vision encoder extraction.

## 2 Related Work

### 2.1 Model Stealing

Model extraction (or model stealing) refers to attacks that replicate the functionality of a deployed victim MLaaS model $f_V$ by querying it as a black box and training a surrogate $f_S$ on the resulting query–response pairs. Early work showed that rich prediction APIs can leak enough information to enable effective extraction under limited assumptions (Tramèr et al., 2016). Subsequent attacks improved practicality for deep models by constructing large transfer sets and distilling the victim to match its outputs (functionality stealing) (Orekondy et al., 2019), and by developing higher-fidelity extraction procedures that better approximate the victim beyond mere task accuracy (Jagielski et al., 2020). Data-free extraction replaces natural queries with synthetic inputs optimized to elicit informative responses for distillation (Truong et al., 2021). The MLaaS deployment of encoders enables model extraction, extending such attacks to representation models (Dziedzic et al., 2022). Following the recent taxonomy (Zhao et al., 2025), our setting is black-box, API-based *functionality extraction* via knowledge distillation (KD), where the attacker learns a surrogate encoder solely from embedding outputs.

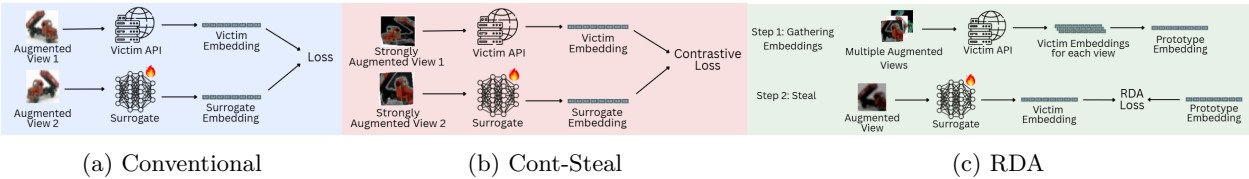

| (a) Conventional | (b) Cont-Steal | (c) RDA |

Figure 1: Conventional and Cont-Steal query the victim online, aligning surrogate embeddings to outputs (light vs. strong augmentations). RDA queries once offline to build prototypes from multiple views, then trains the surrogate to match them.

## 2.2 Encoder Stealing

Encoders are particularly vulnerable to extraction because high-dimensional embeddings expose substantially richer supervision than class-probability APIs, enabling direct representation-level knowledge distillation from black-box queries. Figure 1 summarizes the workflows of three representative API-based attacks that we benchmark. We categorize attacks as *online* or *offline* based on whether they query the victim throughout training (e.g., in each epoch) or query the victim once upfront before training begins. Conventional (Dziedzic et al., 2022) trains a surrogate by matching $f_S(x)$ to $f_V(x)$ using regression losses (e.g., MSE) or contrastive alignment in an InfoNCE-style objective (Dziedzic et al., 2022; Oord et al., 2018; Chen et al., 2020). Cont-Steal (Sha et al., 2023) argues that pointwise matching underutilizes relational information and instead transfers embedding-space geometry via contrastive learning over strongly augmented views, encouraging neighborhood preservation and reducing representation collapse. RDA (Wu et al., 2024) further improves robustness and sample/query efficiency by reducing augmentation-induced variance: it aggregates multiple victim queries per seed into a stable prototype and then trains the surrogate to match these prototypes with objectives that constrain both angular similarity and embedding magnitude. StolenEncoder (Liu et al., 2022) is another augmentation-based matching attack, which we relegate to the appendix because it is consistently comparable to or worse than RDA across our settings (Table 5) and does not affect our main conclusions. Finally, data-free encoder stealing has been explored for self-supervised models using synthetic inputs (Zhang et al., 2024); we do not benchmark this setting because our protocols explicitly study how extraction varies with the relationship between the victim training distribution, the surrogate query distribution, and the evaluation distribution under controlled dataset shifts and multi-source mixtures (Section 4.1).

## 2.3 Defenses

Defenses against model stealing have largely been developed for classifier-based APIs and include watermarking, output perturbation or truncation, and query behavior detection (Adi et al., 2018; Uchida et al., 2017; Juuti et al., 2019; Orekondy et al., 2019). While some of these strategies have been adapted to encoder APIs, recent work shows that they are substantially less effective in this setting: high-dimensional embeddings provide richer supervision, and augmentation-based, multi-view stealing objectives are robust to stochastic output perturbations (Dziedzic et al., 2022; Sha et al., 2023; Wu et al., 2024). In particular, Cont-Steal and RDA demonstrate that perturbation-based defenses such as adding noise, top-$k$ truncation, or feature rounding cannot achieve a favorable trade-off. Reducing attack success typically requires degrading the victim encoder's downstream utility by a similar or greater margin (Sha et al., 2023; Wu et al., 2024). Moreover, watermarking via backdoors is not reliably preserved under stealing. Both Cont-Steal and RDA can recover high surrogate utility while substantially reducing watermark rate, undermining ownership verification (Adi et al., 2018; Sha et al., 2023; Wu et al., 2024). As a result, encoder stealing remains difficult to defend using techniques designed for classifier outputs.

# 3 Background & Preliminaries

An encoder is a function $f : \mathcal{X} \to \mathbb{R}^d$ that maps an input $x \in \mathcal{X}$ (e.g., an image) to a $d$-dimensional embedding $z = f(x)$. These embeddings act as reusable feature representations and are commonly used as

fixed backbones for downstream tasks such as classification, detection, and retrieval. We consider a *victim* encoder $f_V$ deployed via an API and a *surrogate* encoder $f_S$ trained by an adversary. The attacker interacts with the victim in a black-box manner, observing only the output embeddings $f_V(x)$ for queried inputs $x$, without access to the victim's architecture, parameters, or training procedure. *Encoder stealing* is the task of learning a surrogate encoder whose representations approximate those of the victim by querying the victim on selected inputs. Formally, the attacker uses a query *surrogate* dataset $D_S = \{x_i\}_{i=1}^n$ and collects the embeddings $\{f_V(x_i)\}_{i=1}^n$, which are then used to train $f_S$ by minimizing an attack loss of the form $\min_{f_S} \mathbb{E}_{x \sim D_S} [\mathcal{L}_{\text{attack}}(f_S(x), f_V(x))]$, where $\mathcal{L}_{\text{attack}}$ enforces representation alignment, with different methods defining this loss differently. Existing methods differ in both the data augmentation strategies they employ and how they enforce alignment between $f_S$ and $f_V$. Below, we briefly describe the encoder stealing methods benchmarked in this paper. We focus on Conventional, Cont-Steal, and RDA as representative baseline, contrastive, and distribution-level alignment methods, respectively. We exclude StolenEncoder (Liu et al., 2022), as it primarily augments conventional matching with augmentations and does not introduce fundamentally different alignment principles beyond those captured by the above categories. Detailed formulations of all methods are provided in Appendix A.2.

**Conventional.** The conventional approach (Dziedzic et al., 2022) trains the surrogate to align its embeddings $f_S(x)$ with the corresponding victim embeddings $f_V(x)$ using similarity-based loss functions such as MSE or InfoNCE. Depending on the loss used, this can focus on individual sample alignment or incorporate pairwise relationships, but it does not explicitly model higher-order distribution-level relationships.

**Cont-Steal.** Cont-Steal (Sha et al., 2023) uses a standard contrastive loss objective by using (strongly) augmented views of a given input. It treats surrogate–victim pairs as positives and other samples in the minibatch as negatives, and therefore focuses on preservation of neighborhood geometry rather than only pointwise similarity.

**RDA.** Representation Distribution Alignment (RDA) (Wu et al., 2024) improves robustness to augmentation noise by querying multiple augmented views per input and aggregating them into a prototype representation. The surrogate is then trained to align with this prototype representation, promoting more stable and distribution-level matching instead of single-view alignment.

## 4 Benchmark Methodology

### 4.1 Threat Models

**Attacker Goals.** The attacker's goal is to steal the functionality of the victim encoder $f_V$ by training a surrogate $f_S$ on query–response pairs $(x, f_V(x))$ in a black-box setting. We evaluate two notions of success. First, for task-specific victims, the attacker primarily cares about the surrogate's utility on the downstream task; success is measured by how much of the victim's linear-probe accuracy the surrogate recovers. Second, for foundation-style victims, the attacker aims to steal a surrogate that can be used as a reliable backbone under distribution shifts, replicating the victim's transferability. In this case, success is measured by the utility across multiple evaluation distributions, including those not used for stealing.

**Attacker Knowledge.** We assume a strict black-box setting consistent with MLaaS services: the attacker observes only the output embedding $z = f_V(x)$ and has no access to the weights, intermediate activations, or gradients. The attacker can infer the embedding dimension $d$ from an API response and use it to configure the surrogate's projection head. Let $\mathcal{D}_V$, $P_V$, and $\mathcal{D}_S$ denote the victim training set, victim training distribution, and attacker's query set, respectively. We consider three regimes to disentangle extraction difficulty from distribution shift: (i) dataset-aware ($\mathcal{D}_S = \mathcal{D}_V$), (ii) distribution-aware ($\mathcal{D}_S \sim P_V$ with $\mathcal{D}_S \cap \mathcal{D}_V = \emptyset$), and (iii) data-agnostic ($\mathcal{D}_S \not\sim P_V$).

**Attacker Capabilities.** The attacker can query the victim, record embeddings, and train a surrogate. Unless stated otherwise, we fix the *unique seed data* budget to $|\mathcal{D}_S| = 9{,}000$. We focus on this constraint because, in practice, acquiring or curating a suitable query set is often the bottleneck; the total number of API queries $Q$ then depends on the attack method. Augmentation-driven attacks (Conventional and Cont-

Steal) query the victim throughout training, while prototype-based methods (RDA) precompute embedding targets using a fixed number of views per sample and then train without further API call (Dziedzic et al., 2022; Sha et al., 2023; Wu et al., 2024). We do not assume the attacker can pre-train an SSL encoder from scratch at comparable cost; extraction is motivated precisely by this gap. For the cross-modal stealing setup (Section 4.4), we assume that the attacker can access the text encoder corresponding to the victim's vision encoder of the vision–language model (VLM).

## 4.2 Distribution Shift Regimes

To steal surrogates under distribution shifts and evaluate them, we instantiate the three knowledge regimes from Section 4.1 through concrete victim–query pairings and evaluate on both the *surrogate* dataset $D_S$ and the *victim* dataset $D_V$. Evaluating on $D_S$ measures attack success on the queried distribution, capturing how well the surrogate matches the victim on the inputs the adversary actually uses for extraction and is most relevant when the surrogate will be deployed on similarly sourced data. Evaluating on $D_V$ measures utility to the victim's training distribution, which reveals whether the stolen encoder transfers under distribution shift and quantifies how much training-distribution knowledge is required to replicate the victim's utility. Reporting both separates attacks that only achieve local imitation on $D_S$ from those that recover representations that generalize to $D_V$.

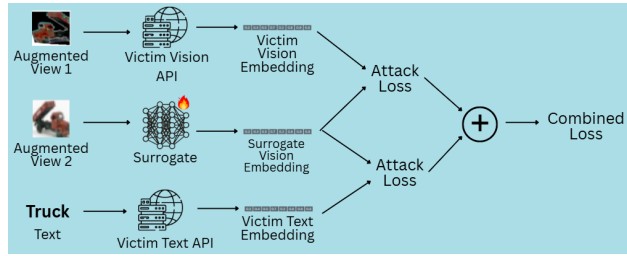

Figure 2: Cross-Modal Text Guidance for VLM victims. The surrogate is aligned to both modalities using a composite objective over their attack losses.

In the dataset-aware regime ($\mathcal{D}_S = \mathcal{D}_V$), we query task-specific victims using unseen datapoints from their own training dataset. This setting is an empirical upper bound: it isolates distillation loss from distribution shift by evaluating the surrogate on the victim's native test split.

In the distribution-aware regime ($\mathcal{D}_S \sim P_V$ with $\mathcal{D}_S \cap \mathcal{D}_V = \emptyset$), we use semantically overlapping dataset pairs (e.g., CIFAR-10↔STL-10). We evaluate both (i) *local utility* on the query dataset and (ii) *transfer back* to the victim dataset, which tests whether stealing preserves the representation in a way that remains useful on the victim's intended distribution.

In the data-agnostic regime ($\mathcal{D}_S \not\sim P_V$), we enforce a distribution mismatch. Here we again evaluate utility on (i) the query dataset, capturing how well the surrogate adapts to what it actually sees, and (ii) the victim dataset, capturing whether any of the victim's structure is recovered despite the mismatch. For foundation victims with proprietary pretraining corpora, this regime is the default in practice.

## 4.3 Multi-Source Data Aggregation

To build a generalizable surrogate across datasets, we query foundation-model victims with samples from multiple sources, since these models are the most realistic EaaS targets and their broad representations enable cross-distribution transfer. We hypothesize that semantic coverage matters more than strict distribution matching, though excessive diversity may dilute density. With a fixed seed budget of $|\mathcal{D}_S| = 9,000$, we study how surrogate-set diversity impacts extraction by mixing datasets of varying overlap, and evaluate transfer on each constituent dataset. Concretely, we compare three mixture protocols: (i) *homogeneous*, mixing closely matched sources to introduce controlled variability; (ii) *heterogeneous*, mixing dissimilar sources to construct a cross-distribution query set; and (iii) *global*, combining all sources to maximize diversity under the fixed budget. In all cases, each constituent subset is class-stratified, contribute equal samples, and is shuffled to avoid curriculum bias.

Table 1: Linear-probe top-1 accuracy (%) of victims across datasets. Task-specific ResNet-34s train on one dataset and evaluated on all three; foundation models excel on natural images but lag on structured digits (SVHN). Bold indicates the best model for each dataset.

| Victim Architecture | Train Dataset | Eval Dataset | | |
|---|---|---|---|---|
| | | CIFAR-10 | STL-10 | SVHN |
| *Task Specific Models* | | | | |
| ResNet-34 | CIFAR-10 | 80.66 | 67.85 | 64.79 |
| ResNet-34 | STL-10 | 79.34 | 72.16 | 56.84 |
| ResNet-34 | SVHN | 57.36 | 51.89 | **80.53** |
| *Foundation Models* | | | | |
| ViT-B/16 (CLIP) | WebImageText | **95.27** | 99.03 | 67.99 |
| ViT-B/16 (DINO) | ImageNet | 95.16 | **99.15** | 70.78 |

### 4.4 Cross-Modal Text Guidance

For multi-modal models, we hypothesize that the rich semantic structure of their text encoder can serve as a robust guidance for the surrogate. To leverage this, we consider a composite objective that anchors the surrogate to both the visual and textual manifolds of the victim (Figure 2). Formally, for each query image $x_i$, we construct a text prompt $t_i$ from available class-name templates, mimicking a scenario where semantic metadata is available. This assumption is stronger than pure black-box access; in practice, class names could be obtained via a captioning model. The surrogate $f_S$ is then optimized to minimize the divergence from both the victim's visual encoder $f_V$ and the frozen text encoder $E_T$. Let $\mathcal{L}_{attack}(\cdot, \cdot)$ denote the specific loss function employed by the stealing algorithm. We define the final objective as:

$$\mathcal{L}_{total} = \underbrace{\mathcal{L}_{attack}(f_S(x_i), f_V(x_i))}_{\text{Visual Fidelity}} + \underbrace{\mathcal{L}_{attack}(f_S(x_i), E_T(t_i))}_{\text{Semantic Alignment}}. \tag{1}$$

The objective in equation 1 is expressed at the representation level and does not explicitly model attack-specific augmentations or multi-view transformations, which are retained as defined by the underlying stealing method. Here, the first term ensures the surrogate replicates the victim's visual topology, while the second term aligns the surrogate with the victim's text embedding space. Crucially, semantic alignment utilizes the identical loss formulation as the visual attack, ensuring consistent optimization dynamics across modalities.

## 5 Experiments and Results

We structure our experimental evaluation around the benchmark methodology introduced in Section 4. We begin by specifying the victim and surrogate architectures, datasets, and evaluation metrics. Extraction performance is reported using Top-1 accuracy under the linear-probe protocol, and results are organized along four experimental axes: (i) distribution-shift regimes, (ii) multi-source data aggregation, (iii) cross-modal text guidance for CLIP-based victim model, and (iv) data and query efficiency, as well as architecture ablations, for both victim and surrogate models.

### 5.1 Experimental Setup

**Model Architectures.** For task-specific victims, we use ResNet-34 (He et al., 2016) encoders trained with contrastive InfoNCE-style objectives separately on CIFAR-10, STL-10, and SVHN (Dziedzic et al., 2022) for 200 epochs. For foundation-style victims, we target general-purpose encoders (CLIP and DINO) (Radford et al., 2021; Caron et al., 2021). For *victim ablations*, we fix the surrogate architecture to be ResNet-34 and vary the victim models (namely ResNet-34, CLIP, and DINO) to measure cross-architecture transferability. For *surrogate ablations*, we fix the victim model to ResNet-34 and vary the surrogate architectures across

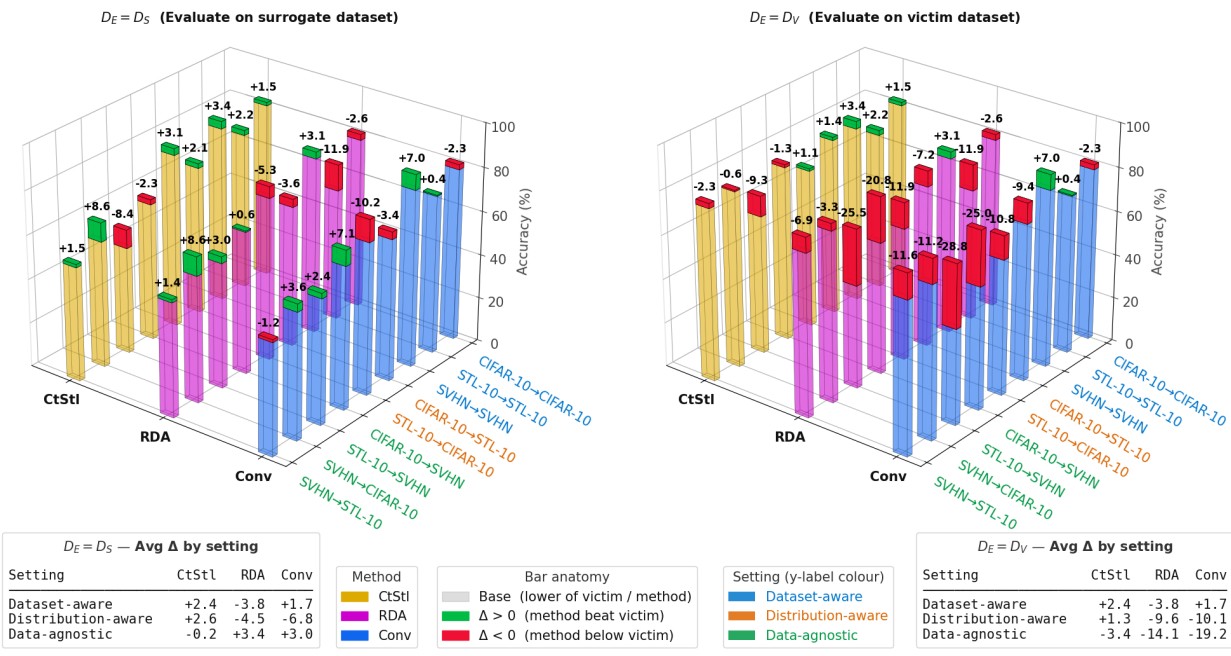

Figure 3: ResNet-34 victims and surrogates under distribution shift. Bar height shows max(victim,surrogate) accuracy; colored tips mark $\Delta = \text{surrogate} - \text{victim}$ (green: $\Delta > 0$, red: $\Delta < 0$). Tables report avg $\Delta$ per setting. CtStl is most robust on $D_V$ while Conv/RDA overfit $D_S$ under severe shift. $D_S$: Surrogate Dataset; $D_V$: Victim Dataset; $D_E$: Evaluation Dataset; CtStl: Cont-Steal; Conv: Conventional.

ResNet-18/34/50 to study sensitivity to surrogate capacity. Surrogate models are trained for 200 epochs using the default attack parameters.

**Datasets.** We instantiate controlled distribution shifts between $\mathcal{D}_S$ and $\mathcal{D}_V$ using CIFAR-10, STL-10 (unlabeled subset), and SVHN (Krizhevsky et al., 2009; Coates et al., 2011; Netzer et al., 2011). CIFAR-10 serves as a canonical natural-image target for task-specific victims. STL-10 provides semantically related natural images with different source and resolution, which we use as a mild distribution shift setting. SVHN is a structured digit dataset with limited semantic overlap with object-centric natural image datasets, and we use it as a strong out-of-distribution (OOD) query source.

**Evaluation Metrics.** We evaluate both victims and stolen surrogates via the *linear evaluation* protocol described by (Dziedzic et al., 2022), i.e., we freeze the encoder and train a linear classifier for 100 epochs, reporting Top-1 accuracy on the chosen evaluation dataset $D_E$.

**Victim Baselines.** Table 1 reports linear-probe Top-1 accuracy for each of the victim encoders. CLIP/DINO substantially outperform the task-specific ResNet-34 on natural images (CIFAR-10/STL-10), while task-specific ResNet-34 trained on SVHN performs best on structured digits (SVHN). This illustrates that while foundation encoders transfer broadly on natural images, their performance can be suboptimal on structured distributions orthogonal to their pretraining distribution. Interestingly, the ResNet-34 trained on STL-10 transfers better to CIFAR-10 than to its own STL-10 test set, likely due to STL-10's higher complexity and variability. Given that the attacker is unaware of the victim's training distribution, these discrepancies underscore the importance of evaluating extraction under controlled distribution shifts.

## 5.2 Distribution Shift Regimes

**Task-specific victims.** Figure 3 and Appendix Table 9 consolidates results across three regimes: dataset-aware (no shift), distribution-aware (mild shift: CIFAR-10↔STL-10), and data-agnostic (severe shift: SVHN↔CIFAR-10/STL-10). We report Top-1 Accuracy on both the stealing dataset $D_S$ and the victim dataset $D_V$. In the dataset-aware setting, all attacks recover most victim utility and can even exceed

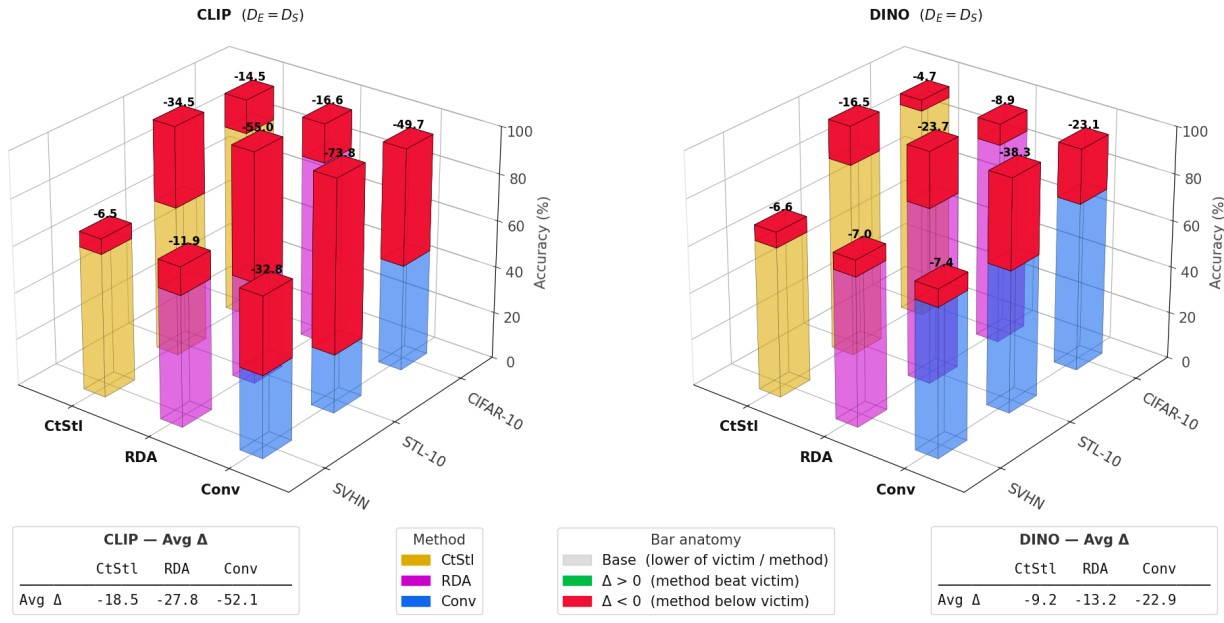

Figure 4: Foundation-model victim extraction, data-agnostic setting (surrogate: ResNet-34), $D_E = D_S$. Base bar height is min(surrogate acc, victim acc); colored tips show $|\Delta|$ where $\Delta$ = surrogate − victim acc (green: $\Delta > 0$, red: $\Delta < 0$). Bottom tables report avg $\Delta$ per encoder. CtStl consistently outperforms RDA and Conv, yet all surrogates remain well below the victim; DINO is more stealable than CLIP. $D_S$: Surrogate Dataset; $D_E$: Evaluation Dataset, CtStl, Cont-Steal; Conv, Conventional.

it. Under the distribution-aware setting, Cont-Steal remains consistently strong on $D_V$ (average $\Delta$ +1.26), while Conventional and RDA degrade more noticeably. Under severe mismatch, Conventional and RDA often fit well to $D_S$ (positive $\Delta$) but lose substantial utility on $D_V$. In contrast, Cont-Steal most reliably preserves victim utility on $D_V$, indicating that instance-discrimination objectives are more robust against distribution mismatch than conventional methods or prototype matching approaches.

**Foundation victims.** As foundation victims have proprietary pretraining corpora, their experiments lie in the data-agnostic regime. Figure 4 and Appendix Table 10 summarizes foundation-victim extraction using a ResNet-34 surrogate. Across all datasets, surrogate accuracy remains well below the victim for both CLIP and DINO, with consistently larger degradation for CLIP. Conventional is particularly weak (e.g., CLIP drops to 25.17 on STL-10 and 35.18 on SVHN), while Cont-Steal is the strongest attack but still incurs sizeable gaps (average $\Delta$: $-18.49$ for CLIP vs. $-9.24$ for DINO). Overall, DINO is more stealable than CLIP under this setting, with Cont-Steal reaching 90.49 on CIFAR-10 for DINO (vs. 80.80 for CLIP) and exhibiting smaller average losses across datasets, whereas RDA lags behind Cont-Steal for both victims.

### 5.3 Multi-Source Aggregation Strategy

To evaluate whether the multi-source strategy in Section 4.3 can yield a *single* surrogate that generalizes across distributions, we steal from foundation victims (CLIP, DINO) using mixed query sets. We fix the unique-seed data budget at $|\mathcal{D}_S| = 9{,}000$ and vary only the composition of the surrogate dataset. For each mixture, we train one balanced surrogate and evaluate it on *all* constituent datasets to measure cross-distribution generalization. Specifically, two-source mixtures allocate 4,500 samples per dataset, while the universal mixture allocates 3,000 per dataset. This setup isolates the effect of dataset diversity under a constant budget and reflects a practical trade-off: training multiple task-specific surrogates versus training a single surrogate that performs well across all distributions.

Figure 5 and Table 11 show that surrogate dataset diversity helps only when it avoids major distributional mismatch. Mixing *natural-image* sources (CIFAR-10+STL-10) is relatively benign: the Cont-Steal surro-

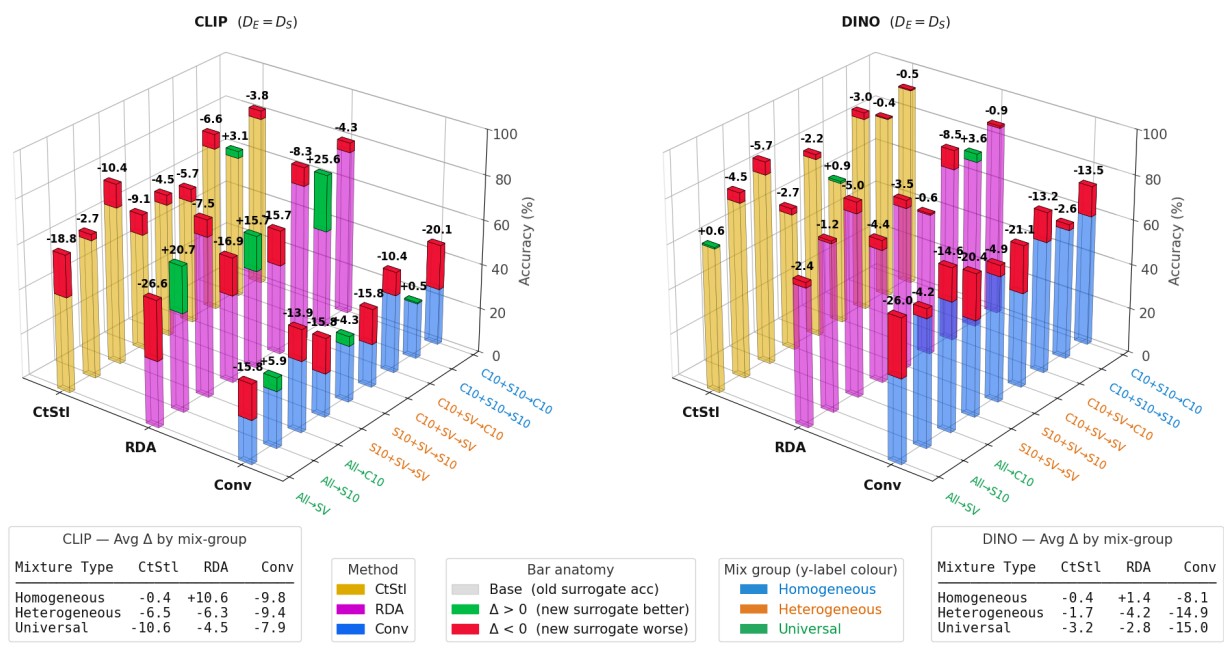

Figure 5: Mixed-source surrogate sets against foundation model victims (CLIP, DINO). Base bar height is min(surrogate acc, old surrogate acc); colored tips show $|\Delta|$ where $\Delta =$ surrogate $-$ old surrogate (green: $\Delta > 0$, red: $\Delta < 0$). Bottom tables report avg $\Delta$ for each mixture type and attack. Semantic alignment helps (CIFAR-10+STL-10); SVHN-inclusive and universal mixes are net-negative, reflecting a density–diversity trade-off. $D_S$: Surrogate Dataset; $D_E$: Evaluation Dataset, CtStl, Cont-Steal; Conv, Conventional.

gate stays close to its single-source baseline (average $\Delta \approx -0.37$ for CLIP and $-0.44$ for DINO), while RDA improves substantially for CLIP (average $\Delta \approx +10.65$). In contrast, adding SVHN degrades performance across attacks (CLIP average $\Delta \approx -9.42/-6.47/-6.31$; DINO $\Delta \approx -14.92/-1.72/-4.24$ for Conventional/Cont-Steal/RDA), and the universal mix remains net-negative (CLIP $\Delta \approx -9.00/-6.50/-1.93$; DINO $\Delta \approx -13.41/-1.92/-2.53$). These results reflect a density-diversity trade-off under a fixed budget: heterogeneous mixes increase distribution shift and reduce per-dataset coverage (3,000 queries each), most visibly hurting CLIP on SVHN. Practically, a single "generalizable" surrogate benefits Cont-Steal and RDA when mixing semantically aligned natural images. For heterogeneous or universal mixes, however, strong cross-distributions utility may require larger budgets or deliberate design, and task-specific surrogates may remain preferable for robustness on challenging targets.

## 5.4 Cross-Modal Semantic Guidance

Because CLIP is trained with an image–text alignment objective, we evaluate whether exposing the attacker to *text-side* supervision improves extraction (Section 4.4). Figure 6 and Appendix Table 12 compares vision-only stealing to a joint vision+text objective across source/evaluation shifts.

Cross-modal guidance consistently strengthens Conventional and Cont-Steal, with gains concentrated on *in-distribution* task setting performance. When CIFAR-10 is the source, adding text yields clear improvements on CIFAR-10 and STL-10 (Conventional: $+18.00/+8.97$; Cont-Steal: $+5.80/+0.62$), while transfer to SVHN changes only minimally (within $\pm1.12$). The largest effect appears when stealing from SVHN: in-distribution SVHN accuracy jumps by $+27.26$ (Conventional) and $+22.35$ (Cont-Steal), indicating that text prompts help the extraction of CLIP's embedding space geometry.

In contrast, RDA does not reliably benefit from text guidance and can degrade substantially, especially for SVHN-sourced stealing (e.g., $-19.41$ in-distribution SVHN and $-8.34/-10.80$ on CIFAR-10/STL-10). We attribute this to objective interference: enforcing cross-modal alignment can conflict with RDA's multi-view consistency matching unless carefully tuned. Overall, semantic guidance provides a strong, low-overhead

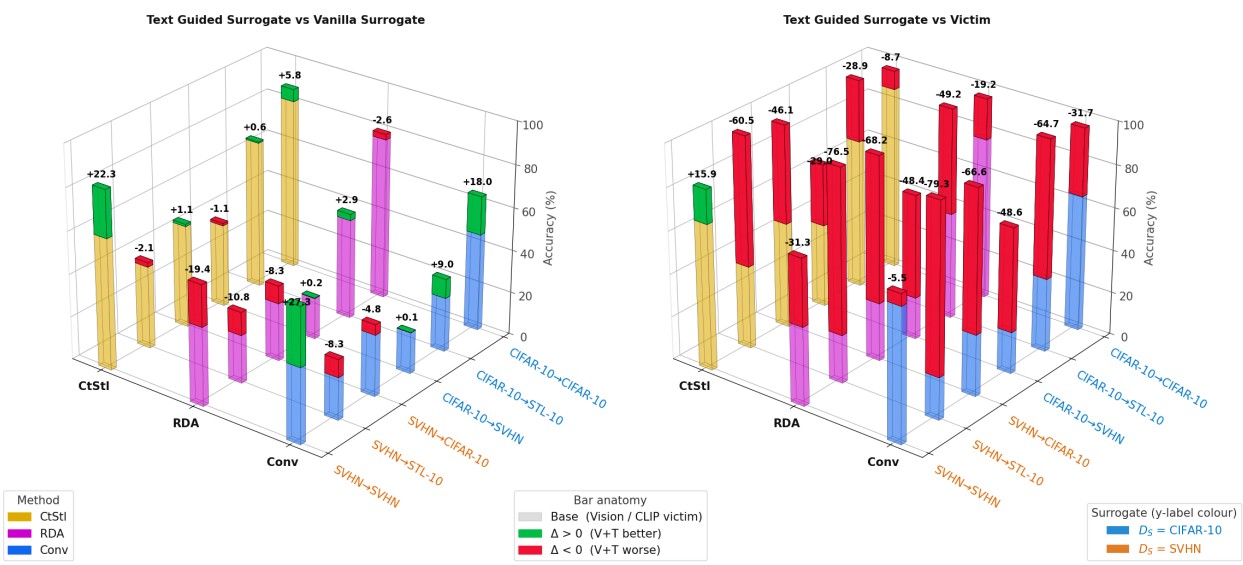

Figure 6: *Left:* $\Delta$ = Vision+Text surrogate $-$ Vision surrogate (effect of adding text guidance). *Right:* $\Delta$ = Vision+Text surrogate $-$ victim (absolute extraction gap). Text guidance consistently boosts Conv and CtStl within-distribution; RDA shows inconsistent or negative gains, suggesting interference with its multi-view prototype alignment. Notably, CtStl with Vision+Text surpasses the victim on SVHN. $D_V$: Victim Dataset; $D_S$: Surrogate Dataset; $D_E$: Evaluation Dataset.

Table 2: Resource trade-off matrix. No single method is universally best; the preferred attack depends on the adversary's bottleneck. We observe a trade-off between query efficiency and local compute (time/TFLOPs): RDA reduces queries but is computationally expensive, whereas Cont-Steal requires more queries but is cheaper to compute. Bold marks the best method for each resource.

| Method | Data Efficiency | Query Efficiency | Stealing Time | Compute Cost | Memory Footprint |
|---|---|---|---|---|---|
| Cont-Steal | **High** (best) | Moderate | Moderate | **Low** ($1.0\times$) | 1.65 GB |
| RDA | Moderate | **High** (best) | High (slowest) | High ($3.8\times$) | 5.20 GB |
| Conventional | Low | Moderate | **Low** (fastest) | **Low** ($1.0\times$) | **1.37 GB** |

improvement for extracting CLIP vision encoder using Conventional or Cont-Steal attack, largely preserving cross-distributions transfer, while RDA requires more deliberate integration to avoid conflicts between its internal objectives and the added text-alignment signal.

## 5.5 Ablation Studies

We conduct controlled ablations around a default configuration consisting of a ResNet-34 victim and ResNet-34 surrogate on CIFAR-10. For the data-size ablation, we vary the query budget $|\mathcal{D}_S|$ from 256 to 9,000 and report (i) Accuracy, (ii) wall-clock training time (seconds), (iii) peak and average GPU memory consumption (GB), and (iv) total training compute (in TFLOPS). This helps characterize how extraction quality and resource requirements scale with data availability. For the model-size ablation, we fix CIFAR-10 and $|\mathcal{D}_S| = 9,000$, then vary the surrogate architecture (ResNet-18/34/50) to study the effect of surrogate capacity. We also vary the victim architecture (ResNet-34, CLIP, DINO) to compare extraction behavior across task-specific and foundation-style encoders.

**Resource Ablations.** Beyond accuracy, the attacks occupy different operating points in the adversary's resource space. Table 2 summarizes the findings of Appendix B.1 (Data Budget Scaling) and Appendix B.2 (Operational Cost) into a single resource trade-off view. Cont-Steal is the strongest choice when the bottleneck is data efficiency, achieving the best utility at low budgets while keeping local compute comparable

Table 3: Victim model ablation (Surrogate: ResNet-34). Accuracy on CIFAR-10. Parentheses show $\Delta = $ (surrogate acc) $-$ (victim acc) for the same victim. Cont-Steal transfers best across the CNN–ViT gap.

| Victim | Victim Acc. | Surrogate Acc. | | |
| --- | --- | --- | --- | --- |
| | | Conventional | Cont-Steal | RDA |
| RN34 | 80.66 | 78.39 (-2.27) | 82.19 (+1.53) | 78.01 (-2.65) |
| CLIP (vision) | 95.27 | 45.55 (-49.73) | 80.80 (-14.47) | 78.65 (-16.62) |
| CLIP (vision+text) | 95.27 | 63.55 (-31.72) | 86.60 (-8.67) | 76.09 (-19.18) |
| DINO | 95.16 | 72.05 (-23.11) | 90.49 (-4.67) | 86.25 (-8.91) |
| Avg. $\Delta$ | – | -26.71 | -6.57 | -11.84 |

Table 4: Surrogate model ablation. Accuracy on CIFAR-10 (Victim: ResNet-34, 80.66%). Parentheses show $\Delta = $ (surrogate acc) $-$ (victim acc). Cont-Steal and RDA are relatively stable where as Conventional is surrogate size dependent. RN18/34/50, ResNet-18/34/50.

| Surrogate Arch. | Conventional | Cont-Steal | RDA |
| --- | --- | --- | --- |
| RN18 | 61.70 (-18.96) | 81.70 (+1.04) | 78.26 (-2.40) |
| RN34 | 78.39 (-2.27) | 82.19 (+1.53) | 78.01 (-2.65) |
| RN50 | 22.32 (-58.34) | 81.39 (+0.73) | 78.67 (-1.99) |
| Avg. $\Delta$ | -26.52 | +1.10 | -2.35 |

to Conventional (1.0× compute, 1.65 GB peak memory), but it is not query-efficient because the victim is queried throughout training. RDA sits at the opposite extreme: it has high query-efficiency, since the victim is only called before training to compute prototype targets, but this shifts the burden to the attacker's hardware, making it the slowest method overall and the most expensive in local resources (3.8× compute, 5.20 GB peak memory). Conventional method is the lightest baseline in terms of runtime and memory (1.37 GB peak memory) and is therefore attractive under tight local budgets, but its lower data efficiency makes it less competitive when the goal is to maximize surrogate utility. Overall, our resource ablations show that no single attack is uniformly best: the preferred method depends on whether the adversary is constrained by API-query cost, available GPU memory/compute, or the amount of data they can acquire.

**Cross-Architecture Transferability and Inductive Bias.** A practical attack should succeed without knowing the victim architecture. Using a fixed RN34 surrogate, we attack a supervised CNN victim (ResNet-34), a language-aligned ViT (Dosovitskiy et al., 2020) victim (CLIP), and a self-supervised ViT victim (DINO). Table 3 shows that Cont-Steal bridges the CNN–ViT gap substantially better than Conventional.

**Surrogate Architecture Sensitivity.** We evaluate how surrogate capacity impacts extraction when both training and evaluation are on CIFAR-10 and the victim is a ResNet-34 encoder. Table 4 reports accuracy. Cont-Steal is consistently robust across surrogate backbones, achieving avg. $\Delta = +1.10$, indicating higher accuracy than victim for RN18/RN34/RN50, suggesting low sensitivity to surrogate capacity. RDA is similarly stable (avg. $\Delta = -2.35$). In contrast, Conventional is highly architecture-dependent and can fail catastrophically (ResNet-50: $\Delta = -58.34$), indicating that Conventional is brittle to optimization and inductive-bias mismatches.

## 6 Limitations

Our benchmark focuses on encoder stealing attacks, emphasizing attack paradigms, extraction behavior under distribution shift, and comparative robustness across representative methods. As a result, several limitations should be noted. First, our analysis is intentionally attack centric. While we discuss the limitations of existing defenses in Section 2.2 to contextualize the threat landscape, we do not claim to provide a comprehensive joint evaluation of attack and defense strategies under our threat model. Second, regard-

ing scope, we restrict our benchmark to three representative attacks, Conventional, Cont-Steal, and RDA, spanning a baseline method, a contrastive based method, and distribution level alignment method. Other recent approaches, including data free stealing (Zhang et al., 2024), are not included in the benchmark as these methods are dataset independent, hence out of scope for our current work. Third, our evaluation is limited to encoders widely used in the encoder stealing literature, namely CLIP and DINO (ViT-B/16) and ResNet 34. Larger or more recent architectures, such as DINOv2 (Oquab et al., 2023) and SigLIP (Zhai et al., 2023), may exhibit different stealability characteristics. Fourth, the controlled distribution shifts are instantiated using CIFAR 10, STL 10, and SVHN, all of which are low resolution datasets. Whether the observed findings, including attack robustness rankings and density diversity trade offs, generalize to higher resolution or domain specific settings such as medical or satellite imagery remains an open question. Finally, cross modal experiments assume access to class names or to an auxiliary classification or captioning model for text prompt construction, and sensitivity to prompt quality is not explored. In addition, following encoder stealing literature surrogate evaluation is restricted to linear probes, without assessing downstream tasks such as retrieval, detection, or few shot transfer.

## 7 Broader Impact

This work examines the practical security risks of vision embedding APIs by analyzing the conditions under which deployed encoders can be replicated through query access. Our objective is not to expand offensive capabilities, but to provide a rigorous empirical understanding of existing vulnerabilities so that model providers and security researchers can better assess real world risk. All evaluated attacks are based on previously published peer reviewed methods; our contribution lies in the benchmark design, comparative evaluation, and systematic analysis of failure modes across threat settings.

A key insight from our study is that high surrogate performance on the attacker's query distribution does not necessarily imply comparable utility on the victim's original training distribution, offering a more nuanced view of the severity of current encoder stealing attacks. Beyond intellectual property concerns, such replication can amplify downstream risks, including adversarial transferability and membership inference. We hope these findings encourage the development of defenses that explicitly account for distribution shift and inspire more resilient embedding API designs.

## 8 Discussion and Conclusion

We presented a benchmark of encoder stealing under realistic constraints, including controlled distribution shifts, fixed data budget, multi-source data aggregation, and architectural uncertainty across task-specific and foundation victims. Strong surrogate utility on the stealing distribution $P_S$ does not imply faithful recovery on the victim distribution $P_V$: under severe mismatch, Conventional and RDA can overfit to $D_S$ while losing substantial utility on $D_V$. In contrast, Cont-Steal most consistently preserves victim-distribution utility across shifts, indicating that transferring relational geometry is more robust than prototype alignment.

For foundation victims in the data-agnostic regime, extraction remains challenging and CLIP is consistently harder to steal than DINO; Cont-Steal is the strongest attack, while Conventional is particularly weak. Under a fixed global budget, multi-dataset stealing can improve cross-distribution coverage but typically reduces per-distribution accuracy, exposing a data density versus diversity trade-off. For CLIP, text-guided stealing provides a boost for Conventional and Cont-Steal, with gains concentrated on the in-distribution task setting, while similar integration can degrade RDA. Complementing these task-level trends, embedding-space analyses show that stolen surrogates are consistently closer to the victim than a benign encoder trained on the same data, with Cont-Steal achieving the strongest representation alignment.

Finally, ablations clarify practical operating points: Cont-Steal is most sample-efficient and robust to surrogate capacity, whereas RDA is query-efficient but shifts cost to local compute and memory and remains most expensive overall. The Conventional method is less competitive but also the lightest, making it attractive only under tight resource budgets. Together, our results map when encoder stealing succeeds and which attack are preferred under data, query, and compute constraints, motivating shift-aware evaluations and defenses tailored to embedding APIs.

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

# A Appendix

## A.1 Experimental Setup

All experiments were run on NVIDIA A100 GPUs (20 GB VRAM). Unless stated otherwise, we train each surrogate encoder for 200 epochs with batch size 128 under a fixed *data budget* of 9,000 images. We follow the evaluation hygiene and non-overlap guidance established in Dziedzic et al. (2022). When the surrogate dataset matches the victim dataset ($D_S = D_V$), we enforce a strict non-overlap protocol by sourcing the surrogate data exclusively from the test split (rather than the train split). In this case, the last 1,000 images are reserved from the relevant test split as a fixed evaluation set and draw the surrogate training data from the remainder. After holding out 1,000 images for evaluation, 9,000 images from the remaining pool is randomly sampled to form the surrogate dataset. The only exception is when $D_S = D_V = $ STL10: the STL10 test split contains 8,000 images, so after holding out 1,000, the effective training budget is 7,000. Following Dziedzic et al. (2022), when the embedding dimensions of the victim and surrogate differ, additional layers are appended to the surrogate backbone to align its output dimensionality with that of the victim; otherwise, only the backbone architecture is used.

We implement each attack using the defaults hyperparameters, and only override the ones needed for a controlled comparison (200 epochs, batch size 128, and the fixed data budget). For *StolenEncoder*, we train using weight ($\lambda = 9.0$) and the default augmentation-consistency term, as well as the recommended attacker-side augmentations and optimization settings (e.g., learning rate $10^{-3}$ in the default configuration) (Liu et al., 2022). For *Cont-Steal*, we keep the method's default optimization and augmentation strategy (including RandAugment) under the same global training controls (Sha et al., 2023). For *RDA*, we preserve the method-specific refinement and alignment hyperparameters (including its default temperature and loss weights) while standardizing the surrogate training schedule to 200 epochs for consistency with the rest of the benchmark. Finally, we do not change the checkpointing strategy of the attacks, Conventional, Cont-Steal and StolenEncoder save the latest checkpoint where as RDA saves the best.

## A.2 Attack Formulations

**Conventional.** Conventional method performs per-sample representation matching by training the surrogate to align its embeddings with those of the victim. Given an input $x$, the surrogate minimizes

$$\mathcal{L}_{\text{DE}}(x) = \mathcal{L}_{\text{align}}\big(f_S(x), f_V(x)\big), \tag{2}$$

which may be different distance or similarity measures (e.g., $\ell_2$, cosine). In our experiments, we use an InfoNCE loss, which prior work has shown to yield strong performance (Dziedzic et al., 2022).

**StolenEncoder.** StolenEncoder aligns surrogate and victim representations under augmentations. For an input $x$, the attacker samples two augmentations $t_1(\cdot)$ and $t_2(\cdot)$ and defines

$$
\begin{aligned}
\mathcal{L}_1 &= d\big(f_V(x), f_S(x)\big), \\
\mathcal{L}_2 &= d\big(f_V(t_1(x)), f_S(t_2(x))\big),
\end{aligned}
\tag{3}
$$

where $d$ is a distance metric (e.g., $\ell_2$). The final loss is

$$
\mathcal{L}_{\text{SE}}(x) = \mathcal{L}_1 + \lambda \mathcal{L}_2.
\tag{4}
$$

**Cont-Steal.** Given two augmentations $t_1(\cdot)$ and $t_2(\cdot)$ of an input $x$, the surrogate is trained using a contrastive objective that treats $(f_S(t_1(x)), f_V(t_2(x)))$ as a positive pair and embeddings of other samples in the minibatch as negatives.

$$
\mathcal{L}_{\text{Cont}}(x) = -\log \frac{\exp(\text{sim}(f_S(t_1(x)), f_V(t_2(x)))/\tau)}{\sum_{x' \in \mathcal{B}} \exp(\text{sim}(f_S(t_1(x)), f_V(t_2(x')))/\tau)},
$$

where $\text{sim}(\cdot, \cdot)$ denotes cosine similarity, $\tau$ is a temperature parameter, and $\mathcal{B}$ is a minibatch.

**RDA.** RDA reduces noise from augmentations by aggregating multiple victim embeddings per sample into a stable prototype (Wu et al., 2024). For each input $x$, the attacker queries $K$ augmented views and computes

$$
\mu_V(x) = \frac{1}{K} \sum_{k=1}^{K} f_V(t_k(x)).
\tag{5}
$$

The surrogate is trained to align its output with this prototype using objectives that jointly constrain angular similarity and embedding magnitude:

$$
\mathcal{L}_{\text{RDA}}(x) = \mathcal{L}_{\text{angle}}\big(f_S(x), \mu_V(x)\big) + \mathcal{L}_{\text{norm}}\big(f_S(x), \mu_V(x)\big).
\tag{6}
$$

RDA aims to more robustly match the victim's embedding distribution than direct pointwise matching by aligning with sample-wise prototypes.

### A.3 StolenEncoder Experiments

In the main paper, we focus on a compact set of encoder-stealing baselines that are both widely used and representative of modern black-box extraction objectives: baseline (*Conventional*) (Dziedzic et al., 2022), contrastive objective stealing (*Cont-Steal*) (Sha et al., 2023), and query-efficient prototype-based extraction (*RDA*) (Wu et al., 2024). Cont-Steal and RDA are especially relevant to our setting because they were shown to be data efficient and query efficient respectively (Sha et al., 2023; Wu et al., 2024).

*StolenEncoder* (Liu et al., 2022) is an important prior method that also performs embedding matching with data augmentation. However, it does not change our conclusions: across dataset-aware, distribution-aware, and data-agnostic regimes, it is generally comparable to or slightly below RDA across our settings (Table 5), with occasional marginal gains. Including it in the main text would therefore add experimental and methodological overhead without providing additional insight for our central questions (shift robustness, multi-source generalization, and semantic guidance). We report StolenEncoder in the appendix for completeness and reproducibility.

### A.4 Additional Experiments

#### A.4.1 Distribution Shift Regimes using ResNet-18 Surrogate

**Task-specific victims.** Appendix Table 6 repeats the evaluation protocol from Section 4.2 using a ResNet-18 surrogate architecture, while keeping the same ResNet-34 victim explored in Section 5.2. The results

Table 5: StolenEncoder vs. RDA under dataset-aware, distribution-aware, and data-agnostic settings. Accuracy (%) is reported for $D_E = D_S$ and $D_E = D_V$. Parentheses next to StolenEnc show gain over RDA: $\Delta = (\text{StolenEnc}) - (\text{RDA})$ (green positive, red negative). We report average $\Delta$ per column within each group and overall. StolenEncoder underperforms RDA in most cases. StolenEnc, StolenEncoder; $D_V$, Victim Dataset; $D_S$, Surrogate Dataset; $D_E$, Evaluation Dataset.

| $D_V$ | $D_S$ | $D_E = D_S$ | | $D_E = D_V$ | |
|---|---|---|---|---|---|
| | | RDA | StolenEnc | RDA | StolenEnc |
| *Dataset-aware* | | | | | |
| CIFAR-10 | CIFAR-10 | 78.01 | 77.62 (-0.39) | 78.01 | 77.62 (-0.39) |
| STL-10 | STL-10 | 60.21 | 59.37 (-0.84) | 60.21 | 59.37 (-0.84) |
| SVHN | SVHN | 83.61 | 84.43 (+0.82) | 83.61 | 84.43 (+0.82) |
| Avg. gain ($\Delta$) | | – | -0.14 | – | -0.14 |
| *Distribution-aware* | | | | | |
| CIFAR-10 | STL-10 | 64.21 | 63.63 (-0.59) | 73.48 | 75.82 (+2.34) |
| STL-10 | CIFAR-10 | 74.08 | 73.12 (-0.96) | 60.21 | 61.11 (+0.90) |
| Avg. gain ($\Delta$) | | – | -0.78 | – | +1.62 |
| *Data-agnostic* | | | | | |
| CIFAR-10 | SVHN | 65.35 | 62.83 (-2.52) | 59.84 | 55.86 (-3.98) |
| STL-10 | SVHN | 59.82 | 56.89 (-2.93) | 46.70 | 46.73 (+0.03) |
| SVHN | CIFAR-10 | 65.96 | 61.93 (-4.03) | 77.19 | 70.47 (-6.72) |
| SVHN | STL-10 | 53.24 | 52.77 (-0.47) | 73.59 | 72.01 (-1.58) |
| Avg. gain ($\Delta$) | | – | -2.49 | – | -3.06 |
| Overall Avg. gain ($\Delta$) | | – | -1.29 | – | -1.77 |

Table 6: ResNet-18 surrogates (Conventional/Cont-Steal/RDA) against the same RN34 victim accuracies as Table 1. Accuracy (%) is reported for $D_E = D_S$ and $D_E = D_V$. Parentheses show $\Delta = $ (surrogate)$-$(victim). $D_V$, Victim Dataset; $D_S$, Surrogate Dataset; $D_E$, Evaluation Dataset; Conv, Conventional; CtStl, Cont-Steal.

| $D_V$ | $D_S$ | $D_E = D_S$ | | | | $D_E = D_V$ | | | |
|---|---|---|---|---|---|---|---|---|---|
| | | Victim | Conv | CtStl | RDA | Victim | Conv | CtStl | RDA |
| *Dataset-aware* | | | | | | | | | |
| CIFAR-10 | CIFAR-10 | 80.66 | 61.70 (-18.96) | 81.70 (+1.04) | 78.26 (-2.40) | 80.66 | 61.70 (-18.96) | 81.70 (+1.04) | 78.26 (-2.40) |
| STL-10 | STL-10 | 72.16 | 56.58 (-15.58) | 72.86 (+0.70) | 61.66 (-10.50) | 72.16 | 56.58 (-15.58) | 72.86 (+0.70) | 61.66 (-10.50) |
| SVHN | SVHN | 80.53 | 59.55 (-20.98) | 84.39 (+3.86) | 82.71 (+2.18) | 80.53 | 59.55 (-20.98) | 84.39 (+3.86) | 82.71 (+2.18) |
| Avg. gain ($\Delta$) | | – | -18.51 | +1.87 | -3.57 | – | -18.51 | +1.87 | -3.57 |
| *Distribution-aware* | | | | | | | | | |
| CIFAR-10 | STL-10 | 67.85 | 50.90 (-16.95) | 69.70 (+1.85) | 66.35 (-1.50) | 80.66 | 58.93 (-21.73) | 80.70 (+0.04) | 75.60 (-5.06) |
| STL-10 | CIFAR-10 | 79.34 | 59.98 (-19.36) | 82.13 (+2.79) | 73.69 (-5.65) | 72.16 | 52.44 (-19.72) | 71.84 (-0.32) | 60.80 (-11.36) |
| Avg. gain ($\Delta$) | | – | -18.16 | +2.32 | -3.58 | – | -20.73 | -0.14 | -8.21 |
| *Data-agnostic* | | | | | | | | | |
| CIFAR-10 | SVHN | 64.79 | 51.00 (-13.79) | 65.20 (+0.41) | 64.67 (-0.12) | 80.66 | 50.06 (-30.60) | 76.31 (-4.35) | 57.38 (-23.28) |
| STL-10 | SVHN | 56.84 | 44.50 (-12.34) | 46.60 (-10.24) | 56.68 (-0.16) | 72.16 | 42.25 (-29.91) | 54.30 (-17.86) | 46.42 (-25.74) |
| SVHN | CIFAR-10 | 57.36 | 58.38 (+1.02) | 66.43 (+9.07) | 66.82 (+9.46) | 80.53 | 63.65 (-16.88) | 77.42 (-3.11) | 76.19 (-4.34) |
| SVHN | STL-10 | 51.89 | 45.37 (-6.52) | 55.27 (+3.38) | 49.70 (-2.19) | 80.53 | 52.22 (-28.31) | 78.26 (-2.27) | 66.76 (-13.77) |
| Avg. gain ($\Delta$) | | – | -7.91 | +0.66 | +1.75 | – | -26.43 | -6.90 | -16.78 |
| Overall Avg. gain ($\Delta$) | | – | -13.72 | +1.43 | -1.21 | – | -22.52 | -2.47 | -10.47 |

provide additional support for the trends observed in both the distribution-shift and ablation analyses. In particular, its comparison with Appendix Table 9 highlights that reducing surrogate capacity hurts Conventional method exhibiting the largest degradations across both $D_E=D_S$ and $D_E=D_V$ settings. Cont-Steal and RDA remains largely robust to architectural changes. Consistent with the main paper's conclusion, Cont-Steal provides a balanced performance, RDA performs well on $D_S$ but not on $D_V$, and Conventional is architecture dependent.

**Foundation victims.** Table 7 repeats the data-agnostic foundation-victim extraction study using a smaller ResNet-18 surrogate. The main trends from Table 10 persist: CLIP remains harder to steal than DINO, and Cont-Steal is the strongest method across datasets. Although the average gains suggest that ResNet-18 might be a better surrogate architecture compared to ResNet-34 for foundation model, the degradation remains substantial across attacks (ResNet-18 Avg. $\Delta$: CLIP $-12.50$ for Cont-Steal and $-26.11$ for RDA; DINO $-8.50$ for Cont-Steal), highlighting the sensitivity of foundation-model extraction to surrogate architecture. Overall, the ResNet-18 results are consistent with the main paper: Cont-Steal > RDA > Conventional, and DINO is more stealable than CLIP.

Table 7: Foundation-victim extraction in the Data-agnostic setting (surrogate: ResNet-18). Here $D_E = D_S$; we report Top-1 accuracy (%) on $D_E$. Parentheses show $\Delta = (\text{value}) - (\text{victim acc})$ for the same $D_E$; green is positive, red is negative. Victim accuracies are shown explicitly for CLIP and DINO. $D_S$, Surrogate Dataset; $D_E$, Evaluation Dataset, Conv, Conventional; CtStl, Cont-Steal.

| $D_E = D_S$ | CLIP | | | | DINO | | | |
|---|---|---|---|---|---|---|---|---|
| | Victim | Conv | CtStl | RDA | Victim | Conv | CtStl | RDA |
| CIFAR-10 | 95.27 | 54.96 (-40.31) | 84.71 (-10.56) | 74.71 (-20.56) | 95.16 | 72.34 (-22.82) | 91.17 (-3.99) | 86.15 (-9.01) |
| STL-10 | 99.03 | 31.38 (-67.65) | 74.18 (-24.85) | 46.72 (-52.31) | 99.15 | 64.97 (-34.18) | 84.23 (-14.92) | 78.40 (-20.75) |
| SVHN | 67.99 | 52.01 (-15.98) | 65.89 (-2.10) | 62.54 (-5.45) | 70.78 | 66.39 (-4.39) | 64.20 (-6.58) | 63.98 (-6.80) |
| Avg. gain ($\Delta$) [RN18] | – | -41.31 | -12.50 | -26.11 | – | -20.46 | -8.50 | -12.19 |
| Avg. gain ($\Delta$) [RN34] | – | -52.13 | -18.49 | -27.84 | – | -22.93 | -9.24 | -13.19 |

### A.4.2 Representation Alignment

**Embedding distribution alignment.** Task utility alone does not fully characterize whether a surrogate $f_S$ matches the victim encoder $f_V$ in representation space. To assess representation-level fidelity beyond task utility (in terms of accuracy), we compare the victim ($f_V$) and surrogate ($f_S$) embedding distributions using Sliced Wasserstein Distance (SWD). A benign baseline ($f_B$) trained from scratch is included for reference.

Beyond downstream task accuracy, encoder stealing can be viewed as recovering the victim's *representation space*. Two surrogates may achieve similar task performance while differing in how closely their embeddings match the victim's geometry, distribution, and instance-level semantics. To directly assess representation-level fidelity, we analyze how closely the surrogate encoder aligns with the victim encoder in embedding space using distributional and instance-level similarity metrics.

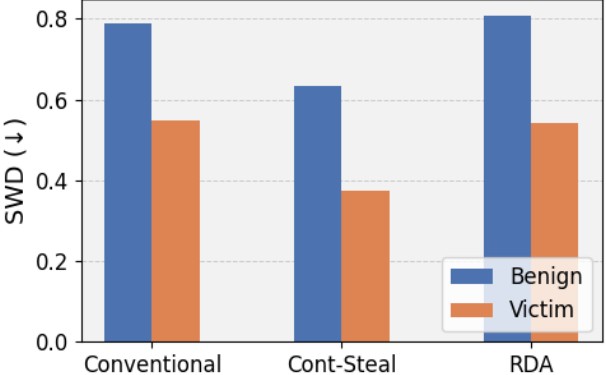

Figure 7: SWD between CLIP-victim and CIFAR-10 test embeddings from stolen surrogates (Conventional, Cont-Steal, RDA) and a benign encoder trained from scratch on CIFAR-10. All stealing methods yield lower SWD than the benign baseline, and Cont-Steal is lowest (closest alignment). Lower mean more aligned.

Let $z_V = f_V(x)$ and $z_S = f_S(x)$ denote embeddings of the same input $x \in \mathcal{D}_E$ produced by the victim and surrogate encoders respectively. We use $\mathcal{D}_E$ to denote a held-out evaluation dataset, disjoint from the surrogate training set, on which representation similarity and downstream utility are measured. Let $P^V$ and $P^S$ be the embedding distributions induced by $f_V$ and $f_S$.

**Sliced Wasserstein Distance (SWD).** To capture geometric differences between embedding distributions, we compute the sliced Wasserstein distance (Kolouri et al., 2018): $\text{SWD}(P^V, P^S) = \mathbb{E}_{\theta \sim S^{d-1}} W_1(\langle Z_V, \theta \rangle, \langle Z_S, \theta \rangle)$, where $W_1$ denotes the 1-Wasserstein distance between one-dimensional projections along random directions $\theta$. SWD emphasizes differences in distributional shape and support, rather than only average similarity.

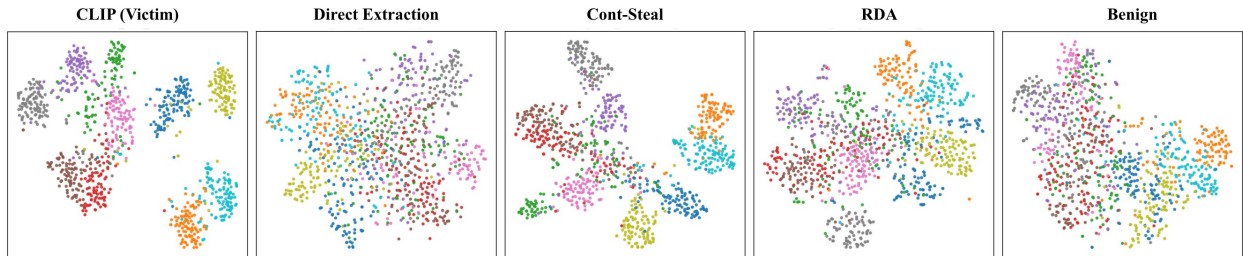

Figure 8: t-SNE of CIFAR-10 test embeddings for the CLIP victim, stolen surrogates (Conventional, Cont-Steal, RDA), and a benign encoder trained from scratch on $\mathcal{D}_S$. Stolen surrogates better preserve the victim's coarse cluster structure than the benign baseline; among surrogates, Cont-Steal exhibits the clearest cluster separation. (CLIP: ViT-B/16; others: ResNet-34.)

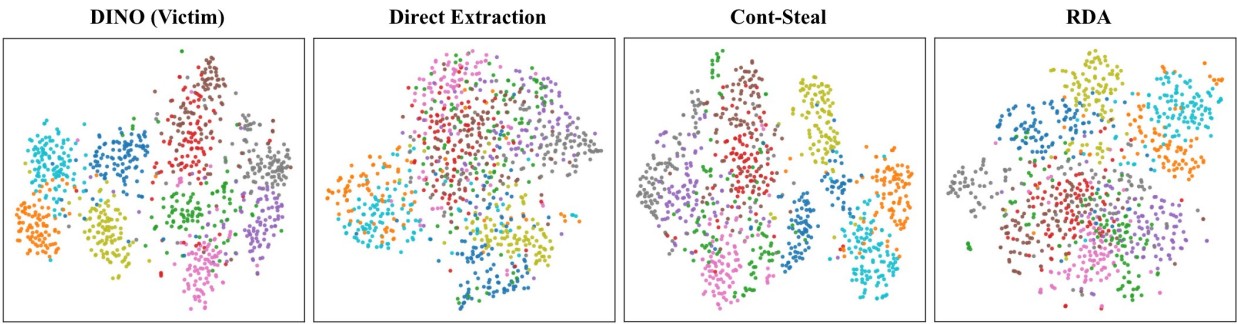

Figure 9: t-SNE visualization of CIFAR-10 evaluation embeddings for the DINO victim encoder and RN34 surrogates (Conventional, Cont-Steal, RDA). Points are colored by CIFAR-10 class.

**Results.** Fig. 7 shows SWD comparisons between victim and surrogate encoders. As a baseline, we also compare stolen encoders with a benign model trained independently on $\mathcal{D}_S$. Across all attacks, surrogate embeddings show consistently lower SWD with respect to the victim compared to the benign encoder, indicating stronger alignment with the victim embedding space. This is expected since surrogate models learn from victim-generated representations, but quantitatively demonstrates that encoder stealing recovers representation structure beyond task-level accuracy. Notably, even when trained on the same data, the benign encoder learns a representation that remains distributionally different from the victim, whereas access to victim representations allows surrogates to inherit aspects of the victim's embedding geometry that are not recovered through data-driven learning alone. Consistent with the accuracy results, Cont-Steal achieves the strongest alignment to the victim among all attacks, suggesting that improved downstream performance is accompanied by closer preservation of embedding geometry. In Fig. 8, we also show t-SNE visualizations of embeddings on the CIFAR10 test set, where clear differences can be observed between the benign trained model and surrogate encoders relative to the victim CLIP model, with Cont-Steal exhibiting the closest structural similarity to the victim representation.

**TSNE plots for DINO.** Figure 9 shows a t-SNE visualization of CIFAR-10 embeddings for the DINO victim alongside ResNet-34 surrogates obtained via Conventional, Cont-Steal, and RDA. The plot suggests that Cont-Steal most closely preserves the victim's coarse clustering and separation patterns, while Conventional produces a more intermixed embedding layout that appears less DINO-like; RDA falls between these two extremes.

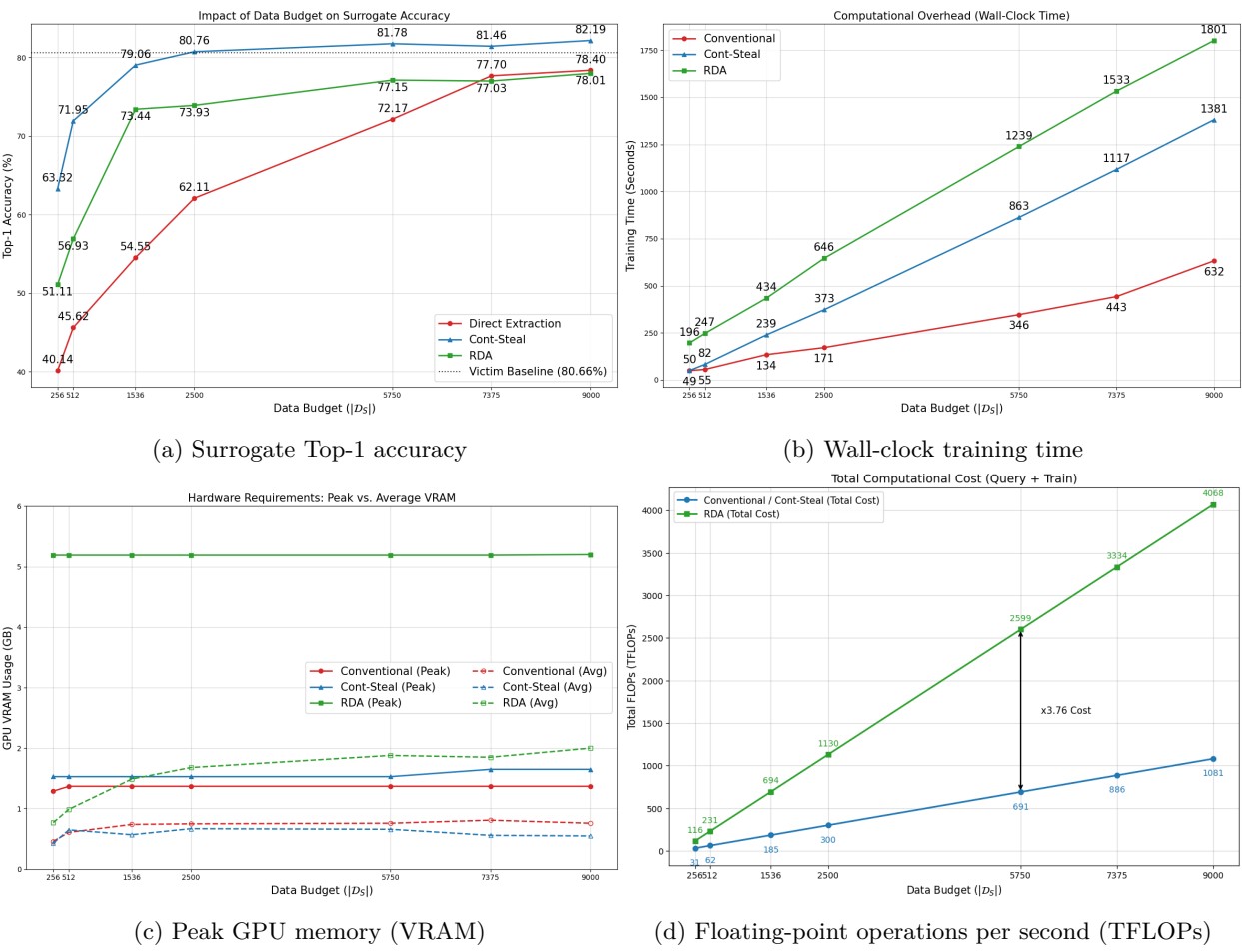

(a) Surrogate Top-1 accuracy

(b) Wall-clock training time

(c) Peak GPU memory (VRAM)

(d) Floating-point operations per second (TFLOPs)

Figure 10: Ablations across stealing methods as a function of surrogate dataset size $|\mathcal{D}_S|$: surrogate Top-1 accuracy, wall-clock training time, peak GPU memory (VRAM), and floating-point operations per second (TFLOPs). Cont-Steal is most data-efficient at low budget; Conventional becomes competitive at larger $|\mathcal{D}_S|$; RDA is the most expensive in time/compute/VRAM due to multiple views and prototypes.

Table 8: Compute cost (TFLOPs) for ablations. Conventional and Cont-Steal query the victim throughout training; RDA queries the victim once (10 views) and trains the surrogate with 5 views per epoch. Total = victim query cost + surrogate training cost. Conv, Conventional; CtStl, Cont-Steal.

| Data Budget ($|D_S|$) | Victim Query (TFLOPs) | | Surrogate Train (TFLOPs) | | Total (TFLOPs) | |
|---|---|---|---|---|---|---|
| | Conv / CtStl | RDA | Conv / CtStl | RDA | Conv / CtStl | RDA |
| 256 | 7.68 | 0.38 | 23.06 | 115.33 | 30.75 | 115.72 |
| 512 | 15.37 | 0.76 | 46.13 | 230.67 | 61.51 | 231.44 |
| 1,536 | 46.13 | 2.30 | 138.40 | 692.02 | 184.54 | 694.33 |
| 2,500 | 75.09 | 3.75 | 225.27 | 1126.35 | 300.36 | 1130.10 |
| 5,750 | 172.70 | 8.63 | 518.12 | 2590.60 | 690.82 | 2599.24 |
| 7,375 | 221.51 | 11.07 | 664.54 | 3322.73 | 886.06 | 3333.80 |
| 9,000 | 270.32 | 13.51 | 810.97 | 4054.86 | 1081.29 | 4068.37 |

## B  Ablations

### B.1  Data Budget Scaling.

**Sample efficiency.** Figure 10a shows accuracy as a function of $|\mathcal{D}_S|$. Cont-Steal is highly sample-efficient, saturating near the victim at $|\mathcal{D}_S| = 2,500$ (80.76%), after which additional data yields diminishing returns. Conventional performs poorly in sparse regimes (40.14% at $|\mathcal{D}_S| = 256$) and requires substantially more data ($|\mathcal{D}_S| = 7375$) to match or surpass RDA.

**Utility per sample vs. utility per query.** Cont-Steal is Pareto-optimal when seed data is scarce (e.g., at $|\mathcal{D}_S| = 256$ it is 12.21 points above RDA), but it is query-intensive because it queries the victim throughout the surrogate's training process. $|\mathcal{D}_S| = 256$, and *epochs* = 200 corresponds to a total of 51,200 queries. RDA trades sample efficiency for query efficiency by querying the victim only before training to construct prototypes, requiring just 2,560 total queries (10 queries/image) at $|\mathcal{D}_S| = 256$. Hence, the optimal attack depends on the adversary's bottleneck: acquiring diverse seed data (favoring Cont-Steal) versus minimizing API call cost (favoring RDA).

### B.2  Operational Cost: Time, Compute, and Memory

We next examine operational costs under varying $|\mathcal{D}_S|$: wall-clock time (Figure 10b), total compute (Figure 10d and Appendix Table 8), and peak GPU memory (Figure 10c).

Query budget approximates monetary cost (API fees), while wall-clock time reflects operational latency and energy. We measure end-to-end training time on a single NVIDIA A100 (20GB). Figure 10b shows a consistent latency ordering: Conventional is fastest; Cont-Steal is $\sim 2\times$ slower in high-budget regimes due to heavier augmentations; RDA is slowest because it processes five views per image during surrogate training (e.g., 196.38s at $|\mathcal{D}_S| = 256$), shifting cost from queries to local compute.

We also measure GPU memory to characterize the hardware barrier. Memory is largely budget-invariant (fixed batch size of 128) and dominated by the algorithmic state. Conventional and Cont-Steal are lightweight (1.37GB and 1.65GB peak VRAM), whereas RDA requires 5.20GB peak ($\sim 3.8\times$ higher), driven by prototype storage and the effective expansion of dataset because of five views per image.

Finally, FLOP accounting further exposes the query-compute trade-off. RDA reduces victim-query FLOPs by 95% relative to Conventional and Cont-Steal (e.g., 0.38 vs. 7.68 TFLOPs at $|\mathcal{D}_S| = 256$), but incurs a $5\times$ higher surrogate-training cost and remains the most expensive overall ($\sim 3.76\times$ higher total TFLOPs). In short, RDA minimizes external cost (API queries) while maximizing internal cost (GPU time/compute), whereas Conventional and Cont-Steal make the opposite trade-off. Therefore, the optimal attack depends on whether the adversary is constrained by API fees or by GPU hours.

### B.2.1 TFLOPs Estimation

We quantify computational overhead in floating-point operations (FLOPs). For a ResNet-34 encoder, profiled via `ptflops`, the per-sample forward pass requires $C_F = 75.09$ MMac $\approx 150.18$ MFLOPs. Following widely adopted convention, the cost of backward pass $C_B$ is estimated as $C_B = 2C_F$, giving a total per-sample training cost of

$$C_T = C_F + C_B = 3C_F. \tag{7}$$

We formalize the total compute complexity for the victim and surrogate models under each attack paradigm as follows. For both the Conventional and Cont-Steal baselines, the victim model performs only forward passes over the surrogate dataset, giving a total victim compute of:

$$\text{TCC}_V^C = C_F \cdot |D_S| \cdot E, \tag{8}$$

where $|D_S|$ is the size of surrogate dataset and $E = 200$ is the number of training epochs. For RDA, which generates multiple augmented views per sample for the victim encoder, this becomes:

$$\text{TCC}_V^{\text{RDA}} = C_F \cdot |D_S| \cdot V_V, \tag{9}$$

where $V_V$ denotes the number of victim-side views per sample.

On the surrogate side, training involves full forward and backward passes, so for the Conventional and Cont-Steal methods:

$$\text{TCC}_S^C = C_T \cdot |D_S| \cdot E = 3 \cdot C_F \cdot |D_S| \cdot E, \tag{10}$$

and for RDA:

$$\text{TCC}_S^{\text{RDA}} = C_T \cdot |D_S| \cdot E \cdot V_S = 3 \cdot C_F \cdot |D_S| \cdot E \cdot V_S, \tag{11}$$

where $V_S$ denotes the number of surrogate-side views per sample. Finally, the total compute complexity for each attack method is obtained by summing the victim and surrogate costs. For the Conventional and Cont-Steal methods:

$$\text{TCC}^C = \text{TCC}_V + \text{TCC}_S = C_F \cdot |D_S| \cdot E + 3 \cdot C_F \cdot |D_S| \cdot E = 4 \cdot C_F \cdot |D_S| \cdot E, \tag{12}$$

and for RDA:

$$\text{TCC}^{\text{RDA}} = \text{TCC}_V^{\text{RDA}} + \text{TCC}_S^{\text{RDA}} = C_F \cdot |D_S| \cdot V_V + 3 \cdot C_F \cdot |D_S| \cdot E \cdot V_S. \tag{13}$$

## C  Supplementary Tables

Table 9: Task-specific ResNet-34 victims under data distribution shift: Actual Accuracy on $D_S$ and $D_V$. Parentheses show $\Delta = $ (surrogate acc) $-$ (victim acc) for the same evaluation dataset; green is positive, red is negative. We report average $\Delta$ per column within each group and overall. The Victim columns report the victim model's accuracy on the corresponding evaluation dataset. Across regimes (dataset-aware, mild shift, severe shift), Cont-Steal is most robust on $D_V$, while Conventional/RDA often overfit $D_S$ under severe mismatch. $D_V$, Victim Dataset; $D_S$, Surrogate Dataset; $D_E$, Evaluation Dataset.

| $D_V$ | $D_S$ | $D_E = D_S$ | | | | $D_E = D_V$ | | | |
|---|---|---|---|---|---|---|---|---|---|
| | | Victim | Conv | CtStl | RDA | Victim | Conv | CtStl | RDA |
| *Dataset-aware* | | | | | | | | | |
| CIFAR-10 | CIFAR-10 | 80.66 | 78.40 (-2.26) | 82.19 (+1.53) | 78.01 (-2.65) | 80.66 | 78.40 (-2.26) | 82.19 (+1.53) | 78.01 (-2.65) |
| STL-10 | STL-10 | 72.16 | 72.60 (+0.44) | 74.40 (+2.24) | 60.21 (-11.95) | 72.16 | 72.60 (+0.44) | 74.40 (+2.24) | 60.21 (-11.95) |
| SVHN | SVHN | 80.53 | 87.52 (+6.99) | 83.95 (+3.42) | 83.61 (+3.08) | 80.53 | 87.52 (+6.99) | 83.95 (+3.42) | 83.61 (+3.08) |
| Avg. gain ($\Delta$) | | – | +1.72 | +2.40 | -3.84 | – | +1.72 | +2.40 | -3.84 |
| *Distribution-aware* | | | | | | | | | |
| CIFAR-10 | STL-10 | 67.85 | 64.47 (-3.38) | 69.95 (+2.10) | 64.21 (-3.64) | 80.66 | 71.29 (-9.37) | 82.09 (+1.43) | 73.48 (-7.18) |
| STL-10 | CIFAR-10 | 79.34 | 69.12 (-10.22) | 82.42 (+3.08) | 74.08 (-5.26) | 72.16 | 61.38 (-10.78) | 73.25 (+1.09) | 60.21 (-11.95) |
| Avg. gain ($\Delta$) | | – | -6.80 | +2.59 | -4.45 | – | -10.08 | +1.26 | -9.57 |
| *Data-agnostic* | | | | | | | | | |
| CIFAR-10 | SVHN | 64.79 | 71.88 (+7.09) | 62.50 (-2.29) | 65.35 (+0.56) | 80.66 | 55.68 (-24.98) | 79.36 (-1.30) | 59.84 (-20.82) |
| STL-10 | SVHN | 56.84 | 59.22 (+2.38) | 48.42 (-8.42) | 59.82 (+2.98) | 72.16 | 43.33 (-28.83) | 62.85 (-9.31) | 46.70 (-25.46) |
| SVHN | CIFAR-10 | 57.36 | 60.96 (+3.60) | 65.98 (+8.62) | 65.96 (+8.60) | 80.53 | 69.30 (-11.23) | 79.92 (-0.61) | 77.19 (+3.34) |
| SVHN | STL-10 | 51.89 | 50.66 (-1.23) | 53.34 (+1.45) | 53.24 (+1.35) | 80.53 | 68.95 (-11.58) | 78.24 (-2.29) | 73.59 (-6.94) |
| Avg. gain ($\Delta$) | | – | +2.96 | -0.16 | +3.37 | – | -19.16 | -3.38 | -14.14 |
| Overall Avg. gain ($\Delta$) | | – | +0.38 | +1.30 | -0.77 | – | -10.18 | -0.42 | -9.69 |

Table 10: Foundation-victim extraction in the Data-agnostic setting (surrogate: ResNet-34). Here $D_E = D_S$; we report Top-1 accuracy (%) on $D_E$. Parentheses show $\Delta = (\text{value}) - (\text{victim acc})$ for the same $D_E$; green is positive, red is negative. Victim accuracies are shown explicitly for CLIP and DINO. Cont-Steal consistently outperforms RDA and Conventional, yet all surrogates remain far below the victim; DINO is more stealable than CLIP (smaller Avg. $\Delta$). $D_S$, Surrogate Dataset; $D_E$, Evaluation Dataset, $\Delta$; Average gain.

| $D_E = D_S$ | CLIP | | | | DINO | | | |
|---|---|---|---|---|---|---|---|---|
| | Victim | Conv | CtStl | RDA | Victim | Conv | CtStl | RDA |
| CIFAR-10 | 95.27 | 45.55 (-49.73) | 80.80 (-14.47) | 78.65 (-16.62) | 95.16 | 72.05 (-23.11) | 90.49 (-4.67) | 86.25 (-8.91) |
| STL-10 | 99.03 | 25.17 (-73.86) | 64.51 (-34.52) | 43.99 (-55.04) | 99.15 | 60.84 (-38.31) | 82.69 (-16.46) | 75.47 (-23.68) |
| SVHN | 67.99 | 35.18 (-32.81) | 61.50 (-6.49) | 56.13 (-11.86) | 70.78 | 63.40 (-7.38) | 64.20 (-6.58) | 63.81 (-6.97) |
| Avg. gain ($\Delta$) | – | -52.13 | -18.49 | -27.84 | – | -22.93 | -9.24 | -13.19 |

Table 11: Mixed-source query sets for foundation victims (surrogate: ResNet-34). We report Top-1 accuracy (%) on each evaluation dataset for CLIP and DINO under four surrogate dataset mixtures. Parentheses show the gain of the *new* surrogate over the *old* surrogate for the same victim, attack, and Eval dataset: $\Delta$ = (new surrogate acc) − (old surrogate acc) (green is positive, red is negative). We report average $\Delta$ per column within each group and overall. Diversity helps mainly for semantically aligned natural-image mixes (CIFAR-10+STL-10), while including SVHN or using the universal mix is generally net-negative, reflecting a data density–diversity trade-off. $D_S$, Surrogate Dataset; $D_E$, Evaluation Dataset; C10, CIFAR-10; S10, STL-10; SV, SVHN.

| $D_S$ (Steal Mix) | $D_E$ | CLIP | | | DINO | | |
|---|---|---|---|---|---|---|---|
| | | Conventional | Cont-Steal | RDA | Conventional | Cont-Steal | RDA |
| *Homogeneous mix (CIFAR-10 + STL-10)* | | | | | | | |
| C10 + S10 | C10 | 25.45 (-20.10) | 76.95 (-3.85) | 74.34 (-4.31) | 58.52 (-13.53) | 90.04 (-0.45) | 85.31 (-0.94) |
| | S10 | 25.71 (+0.54) | 67.62 (+3.11) | 69.59 (+25.60) | 58.26 (-2.58) | 82.26 (-0.43) | 79.11 (+3.64) |
| Avg. gain ($\Delta$) | | -9.78 | -0.37 | +10.65 | -8.06 | -0.44 | +1.35 |
| *Heterogeneous mix (Natural + SVHN)* | | | | | | | |
| C10 + SV | C10 | 35.10 (-10.45) | 74.18 (-6.62) | 70.33 (-8.32) | 58.85 (-13.20) | 87.54 (-2.95) | 77.73 (-8.52) |
| | SV | 19.43 (-15.75) | 55.82 (-5.68) | 40.42 (-15.71) | 42.25 (-21.15) | 65.12 (+0.92) | 63.20 (-0.61) |
| S10 + SV | S10 | 29.44 (+4.27) | 60.06 (-4.45) | 59.67 (+15.68) | 55.93 (-4.91) | 80.53 (-2.16) | 71.99 (-3.48) |
| | SV | 19.43 (-15.75) | 52.38 (-9.12) | 39.24 (-16.89) | 42.97 (-20.43) | 61.52 (-2.68) | 59.45 (-4.36) |
| Avg. gain ($\Delta$) | | -9.42 | -6.47 | -6.31 | -14.92 | -1.72 | -4.24 |
| *Universal mix (CIFAR-10 + STL-10 + SVHN)* | | | | | | | |
| C10 + S10 + SV | C10 | 31.60 (-13.95) | 70.37 (-10.43) | 71.19 (-7.46) | 57.42 (-14.63) | 84.80 (-5.69) | 81.25 (-5.00) |
| | S10 | 31.09 (+5.92) | 61.83 (-2.68) | 64.69 (+20.70) | 56.62 (-4.22) | 78.19 (-4.50) | 74.32 (-1.15) |
| | SV | 19.43 (-15.75) | 42.70 (-18.80) | 29.51 (-26.62) | 37.40 (-26.00) | 64.84 (+0.64) | 61.43 (-2.38) |
| Avg. gain ($\Delta$) | | -7.93 | -10.64 | -4.46 | -14.95 | -3.18 | -2.84 |
| Overall Avg. gain ($\Delta$) | | -9.00 | -6.50 | -1.93 | -13.41 | -1.92 | -2.53 |

Table 12: CLIP: Image-only vs. Text-guided stealing (Top-1 accuracy, %). Columns are grouped by attack (Conventional, Cont-Steal, RDA), each with Vision and Vision+Text subcolumns. Parentheses denote the change from Vision to Vision+Text: $\Delta = (\text{vision+text}) - (\text{vision})$ for the same source/eval configuration. Text guidance consistently boosts Conventional and Cont-Steal, with gains concentrated within the distribution, while transfer to SVHN changes minimally; RDA shows inconsistent or negative gains (notably for SVHN source), suggesting objective interference with its multi-view prototype alignment. $D_V$, Victim Dataset; $D_S$, Surrogate Dataset; $D_E$, Evaluation Dataset.

| $D_S$ | $D_E$ | Conventional | | Cont-Steal | | RDA | |
| --- | --- | --- | --- | --- | --- | --- | --- |
| | | Vision | Vision+Text | Vision | Vision+Text | Vision | Vision+Text |
| | CIFAR-10 | 45.55 | 63.55 (+18.00) | 80.80 | 86.60 (+5.80) | 78.65 | 76.09 (-2.59) |
| CIFAR-10 | STL-10 | 25.35 | 34.32 (+8.97) | 69.50 | 70.12 (+0.62) | 46.92 | 49.84 (+2.92) |
| | SVHN | 19.34 | 19.43 (+0.09) | 40.10 | 38.98 (-1.12) | 19.43 | 19.63 (+0.2) |
| | CIFAR-10 | 33.54 | 28.69 (-4.85) | 48.07 | 49.18 (+1.11) | 35.43 | 27.09 (-8.34) |
| SVHN | STL-10 | 28.11 | 19.77 (-8.34) | 40.65 | 38.58 (-2.07) | 33.28 | 22.48 (-10.80) |
| | SVHN | 35.18 | 62.44 (+27.26) | 61.50 | 83.85 (+22.35) | 56.13 | 36.72 (-19.41) |

Table 13: CLIP baseline vs. text-guided stealing (Top-1 accuracy, %). The CLIP baseline depends on the evaluation dataset $D_E$: 95.27 for CIFAR-10, 99.03 for STL-10, and 67.99 for SVHN. Parentheses denote the gap from the CLIP baseline for the same evaluation dataset, $\Delta = (\text{method}) - (\text{CLIP})$. Cont-Steal remains the strongest overall and is the only method to surpass the CLIP baseline on SVHN when $D_S = \text{SVHN}$. $D_S$, Surrogate Dataset; $D_E$, Evaluation Dataset.

| $D_S$ | $D_E$ | CLIP | Conventional | Cont-Steal | RDA |
| --- | --- | --- | --- | --- | --- |
| | CIFAR-10 | 95.27 | 63.55 (-31.72) | 86.60 (-8.67) | 76.09 (-19.18) |
| CIFAR-10 | STL-10 | 99.03 | 34.32 (-64.71) | 70.12 (-28.91) | 49.84 (-49.19) |
| | SVHN | 67.99 | 19.43 (-48.56) | 38.98 (-29.01) | 19.63 (-48.36) |
| | CIFAR-10 | 95.27 | 28.69 (-66.58) | 49.18 (-46.09) | 27.09 (-68.18) |
| SVHN | STL-10 | 99.03 | 19.77 (-79.26) | 38.58 (-60.45) | 22.48 (-76.55) |
| | SVHN | 67.99 | 62.44 (-5.55) | 83.85 (+15.86) | 36.72 (-31.27) |

