# OpenReview forum: "From Queries to Clones: A Systematic Study of Encoder Stealing Attacks"
_TMLR — Under review for TMLR_

### Review · Reviewer_moBN · 2026-05-17

**Summary Of Contributions:**

This paper presents a comprehensive benchmark and comparative study of representative encoder stealing attacks on CLIP and DINO, two widely used encoders. The authors consider three threat scenarios of increasing realism and evaluate a novel setting where the attacker queries multiple datasets to extract a more generalizable surrogate model. A notable strength of this work is its systematic classification of threat scenarios, which characterizes how extraction performance degrades across dataset-aware, distribution-aware, and data-agnostic attacker settings. The empirical results map practical operating points for these attacks and highlight vulnerabilities relevant to foundation models. This systematic approach addresses the lack of uniform evaluation standards in prior research, which frequently studied encoder stealing attacks in isolation under inconsistent configurations. Ultimately, the authors establish a necessary framework for evaluating model extraction under controlled distribution shifts and resource constraints.

**Audience:**

Yes

**Audience Explanation:**

N/A

**Claims And Evidence:**

Yes

**Claims Explanation:**

N/A

**Requested Changes:**

However, the actual scope of the evaluated security threats remains limited. While the benchmark effectively demonstrates that these attacks produce highly capable functional clones, the methodology fails to measure whether the cloning process inadvertently extracts private training data from the victim model. The paper acknowledges that extraction attacks trigger downstream threats like adversarial examples and membership inference, yet the experimental design omits any evaluation of these capabilities. Furthermore, the datasets used to test distribution shifts, such as CIFAR-10 and SVHN, lack the sensitivity required to assess real-world privacy risks. Consequently, although the benchmark proves the effectiveness of these attacks as functional replicators, it leaves the critical issue of privacy memory extraction unresolved. A robust security benchmark must align its evaluation metrics with the privacy risks claimed in its threat model.

---

### Review · Reviewer_ZMa4 · 2026-06-04

**Summary Of Contributions:**

This paper addresses the issue of the foundation encoder being potentially copied in a black-box manner in encoder-as-a-service scenarios. It proposes a systematic benchmark framework and demonstrates, through various threat regimes, query strategies, attack objectives, and resource ablation, that Cont-Steal is currently the most robust stealing method, DINO is more susceptible to being stolen than CLIP, and cross-modal text guidance can further improve the stealing effect of CLIP vision encoder.

**Audience:**

Yes

**Audience Explanation:**

1.The paper is easy to follow.

2.The paper addresses the security risks of copying the embedding API encoder, a topic that is both relevant and important.

3.The experiments covered a variety of attack methods, data distributions, victim models, and resource constraints, resulting in a substantial workload and a relatively complete empirical benchmark.

**Broader Impact Concerns:**

This work demonstrates effective black-box encoder stealing attacks against encoder-as-a-service systems, bringing potential ethical and security risks. The revealed attack paradigms and cross-modal stealing strategies may be misused by malicious parties to illegally replicate proprietary pre-trained encoders, infringing model developers’ intellectual property and compromising the security of cloud AI services.

**Claims And Evidence:**

Yes

**Claims Explanation:**

1.The paper is easy to follow.

2.The paper addresses the security risks of copying the embedding API encoder, a topic that is both relevant and important.

3.The experiments covered a variety of attack methods, data distributions, victim models, and resource constraints, resulting in a substantial workload and a relatively complete empirical benchmark.

**Requested Changes:**

1.The paper claims a realistic black-box setting in which attackers can only access embedding outputs and have no knowledge of the victim model’s architecture, parameters, or training details. However, this threat model becomes less clear in the cross-modal guidance setting, where the attacker is assumed to access the CLIP text encoder. The authors should clarify whether this setting is still considered black-box, or whether it represents a stronger attacker model.

2.In Section 4.4, the cross-modal guidance setting assumes that attackers can construct class-name prompts for query images. This is a stronger assumption than the standard embedding-only black-box setting. The authors should clarify when such semantic information is available and how this assumption affects the interpretation of the results.

3.The paper defines three attacker knowledge regimes in Section 4.1, namely dataset-aware, distribution-aware, and data-agnostic settings. Although the authors provide intuitive criteria for these regimes, the distinction between “similar distribution” and “unrelated images” is still not sufficiently specific. The authors should further explain how they determine whether a query dataset belongs to the distribution-aware or data-agnostic setting.

4.Most tables report single-shot accuracy values without multi-seed statistics, standard deviations, or confidence intervals. This weakens the reliability of the benchmark conclusions, especially when some reported improvements are relatively small. The authors are encouraged to include experiments over multiple random seeds or provide variance analysis to demonstrate the stability of their findings.

---

### Review · Reviewer_7NgS · 2026-07-01

**Summary Of Contributions:**

This paper studies different methods for stealing encoder machine learning models, such as CLIP or DINO. The main motivation is that Encoder-as-a-Service is very useful in practice as it allows to use the same encoder model for many downstream tasks. However, this can facilitate stealing of proprietary EaaS models, as models output high-dimensional vectors and their release provides rich information to the potential adversary.

Three attacks are considered: Conventional, RDA, and Cont-Steal, with an empirical evaluation and ablation study. The work compares the attacks along the data, query, and time efficiencies, and under three regimes (dataset-aware, distribution-aware and data-agnostic). The datasets used for the comparison are CIFAR-10, STL-10, SVNH. Authors also considers multimodal regime, where the attacker has access to a text encoder which might further help to steal the embedding model.

The main takeaway of the paper is that different attack methods are preferable under different resource constraints and there is no single best method. Another takeaway is that using cross-modal information, such as text, significantly improves the attack efficiency for Conventional and Cont-Steal, but not for the RDA method.

**Audience:**

Yes

**Audience Explanation:**

This study is important for researchers who study defenses of machine learning models. In particular, understanding which attacks are more efficient in which settings (data-, query-, or time-constrained) would help to design better defenses.

**Broader Impact Concerns:**

Current discussion in the Broader Impact section is sufficient.

**Claims And Evidence:**

Yes

**Claims Explanation:**

Paper provides an extensive empirical comparison of three common model stealing approaches across various regimes. The separation into different regimes of attacker knowledge is insightful and the ablation study helps to understand the interplay between many considered settings.

See below for the comment regarding cross-modal experiments, which can be explained more clearly. Also, as the authors highlight at the end, the experiments are performed on the smaller datasets, such as CIFAR 10 and STL 10, and it is not clear whether the conclusion will transfer to higher resolution datasets.

**Requested Changes:**

The results on cross-modal guidance are unclear: what was the motivation behind the particular form of joint loss in Eq. (1)? It seems natural to allow some learnable transformation of $f_S(x_i)$ for the alignment with text representation.

Also, at the end of Section A.1 it is written "we do not change the checkpointing strategy of the attacks, Conventional, Cont-Steal and StolenEncoder save the latest checkpoint where as RDA saves the best". Does this have an effect on the comparison, i.e., does RDA has some unaccounted advantage?

In Section A.4.2, in the Results paragraph it is written "Consistent with the accuracy results, Cont-Steal achieves the
strongest alignment to the victim among all attacks". However, on Figure 7 even a benign encoder has a higher alignment in that case. Could you clarify how the conclusion is made based on current Figure 7 and how does one accounts for the stronger alignment also in the benign case.